# Structure and Thermodynamics of Silicon Oxycarbide Polymer-Derived Ceramics with and without Mixed-Bonding [note 1]

**DOI:** 10.3390/ma14154075

**Published:** 2021-07-22

**Authors:** Casey Sugie, Alexandra Navrotsky, Stefan Lauterbach, Hans-Joachim Kleebe, Gabriela Mera

**Affiliations:** 1Department of Chemistry, University of California Davis, Davis, CA 95616, USA; csugie@ucdavis.edu; 2Peter A. Rock Thermochemistry Laboratory and NEAT ORU, University of California Davis, Davis, CA 95616, USA; Alexandra.Navrotsky@asu.edu; 3Materials of the Universe, School of Molecular Sciences, Arizona State University, Tempe, AZ 851281, USA; 4Institut für Angewandte Geowissenschaften, Technische Universität Darmstadt, Schnittspahnstraße 9, D-64287 Darmstadt, Germany; stefan.lauterbach@geo.tu-darmstadt.de (S.L.); kleebe@geo.tu-darmstadt.de (H.-J.K.); 5Institut für Materialwissenschaft, Technische Universität Darmstadt, Otto-Berndt-Straße 3, D-64287 Darmstadt, Germany

**Keywords:** silicon oxycarbide, polymer-derived ceramics, nanodomain structure, carbon nanorolls, polysiloxanes, energetics

## Abstract

Silicon oxycarbides synthesized through a conventional polymeric route show characteristic nanodomains that consist of sp^2^ hybridized carbon, tetrahedrally coordinated SiO_4,_ and tetrahedrally coordinated silicon with carbon substitution for oxygen, called “mixed bonds.” Here we synthesize two preceramic polymers possessing both phenyl substituents as unique organic groups. In one precursor, the phenyl group is directly bonded to silicon, resulting in a SiOC polymer-derived ceramic (PDC) with mixed bonding. In the other precursor, the phenyl group is bonded to the silicon through Si-O-C bridges, which results in a SiOC PDC without mixed bonding. Radial breathing-like mode bands in the Raman spectra reveal that SiOC PDCs contain carbon nanoscrolls with spiral-like rolled-up geometry and open edges at the ends of their structure. Calorimetric measurements of the heat of dissolution in a molten salt solvent show that the SiOC PDCs with mixed bonding have negative enthalpies of formation with respect to crystalline components (silicon carbide, cristobalite, and graphite) and are more thermodynamically stable than those without. The heats of formation from crystalline SiO_2_, SiC, and C of SiOC PDCs without mixed bonding are close to zero and depend on the pyrolysis temperature. Solid state MAS NMR confirms the presence or absence of mixed bonding and further shows that, without mixed bonding, terminal hydroxyls are bound to some of the Si-O tetrahedra. This study indicates that mixed bonding, along with additional factors, such as the presence of terminal hydroxyl groups, contributes to the thermodynamic stability of SiOC PDCs.

## 1. Introduction

Silicon oxycarbide (SiOC) polymer derived ceramics (PDCs) are a class of advanced silica based ceramic materials that are synthesized through a polymeric route. PDCs have greatly impacted technological and scientific advances in ceramic materials1. They are able to resist crystallization, decomposition, phase separation and creep even at temperatures between 1000 and 1500 °C [1,2,3], have excellent mechanical properties [4], and are biocompatible and bioactive [5,6]. Silicon oxycarbide (SiOC) PDCs are commonly synthesized from precursors such as polysiloxanes and polysilsequioxanes [1,3,7,8,9,10]. SiOC PDCs are X-ray amorphous up to temperatures of approximately 1300 °C, making the structure and energetics of the disordered state very interesting. Despite being X-ray amorphous, PDCs contain distinct domains at the nanoscale, [11,12,13,14] making such microstructure largely responsible for thermodynamic and kinetic stability, properties that contribute to the materials’ resistance to crystallization and oxidation [15].

The nanodomains most commonly consist of graphitic or turbostratic sp^2^ hybridized carbon, called ‘free carbon’, a region rich in silica, and some interfacial bonding region where mixed-bonding (SiO_x_C_4−x_, 0 ≤ x ≤ 4) is largely found. Solid state magic angle spinning nuclear magnetic resonance (MAS NMR) provides a powerful tool for determining local bonding and heterogeneity present at the nanoscale [11,14,16]. Information about the presence of mixed bonding (SiO_x_C_4−x_, 0 ≤ x ≤ 4) that occurs at the interfacial region between the silica rich region and the “free carbon” region is gained from both ^13^C and ^29^Si MAS NMR. Furthermore, Raman spectroscopy helps to identify the sp^2^ hybridized carbon species in the “free carbon” phase. The local nanostructural characteristics of PDCs are investigated by high-resolution transmission electron microscopy [7].

Understanding the thermodynamic stability of such SiOC PDCs, namely the enthalpy of formation from crystalline components, provides valuable information about the properties that make them promising materials for industry, specifically their stability at high temperatures. High temperature oxide melt solution calorimetry is a well-developed calorimetric technique [17,18,19,20] that has been used extensively for thermodynamic studies of PDCs [7,12,13,15,21,22]. Most of these calorimetric studies on SiOC PDCs have found that the enthalpy of formation from crystalline components is negative, meaning they are thermodynamically stable as compared to their crystalline components, with likely positive entropies of formation reflecting disorder and domain structure contributing to further stabilization. A look at their nanostructure provides some insight in the differences observed in their thermodynamic stabilities [1,7], which heavily rely on the compositional and structural make-up of the PDCs.

Moreover, there are reports showing that the interfaces between nanodomains may play an essential role in the thermodynamic stability of PDCs [15,16]. While there has been numerous research on the thermodynamic stability of SiOC PDCs with mixed bonding, only little is known about the thermodynamic stability of SiOC PDCs without mixed bonding. One study that investigated the thermodynamic stability of SiOC PDCs without mixed bonding showed that the enthalpy of formation from crystalline components was slightly negative [22]. A similar study involved two SiCN PDCs, with one that contained mixed bonding at the interface between nanodomains, while the other did not [16]. For the PDC without mixed bonding, the model suggests that there are two distinct phases, Si_3_N_4_ and graphitic “free” carbon, with little to no Si-C bonds. While both samples revealed negative enthalpies of formation from crystalline components, the sample with interfacial bonding was significantly more thermodynamically stable (higher negative enthalpies). In addition, there seems to be a relationship to the hydrogen content of the material, being higher in the sample with interfacial bonding. It was concluded that hydrogen may also contribute to the thermodynamic stability of PDCs.

SiOC PDCs are synthesized by pyrolyzing polysiloxanes and polysilsequioxanes in an inert environment. While the resulting ceramic is X-ray amorphous and has no long-range order, short-range order is present in the form of localized nanodomains. This short-range structure is made up of a continuous mass fractal network of corner shared SiO_x_C_4−x_ tetrahedra with some combination of SiO_4,_ SiO_3_C, SiO_2_C_2_, SiOC_3_, and SiC_4_ tetrahedral units [11], termed mixed-bonding. Voids within this network hold sp^2^ hybridized “free” carbon. When pyrolysis of polysiloxanes takes place in a CO_2_ environment, the microstructure differs in that it no longer contains mixed-bonding, but becomes a nanocomposite of silica and sp^2^ carbon [23,24]. Designing a polymer with sol-gel synthesis that has no Si-C bonds, but does contain carbon groups bonded to the Si through Si-O-C bridges also results in a ceramic with no mixed-bonding [22]. In this case, the carbon phase was observed as multiwall carbon nanotube-like sp^2^ hybridized carbon within a porous silica network, in-situ formed during pyrolysis without the need for any metal catalyst.

This study focuses on two different SiOC ceramics, one of which contains mixed bonding, while the other does not. Both SiOCs were synthesized using the sol-gel route at room temperature in water, which is advantageous as there is no requirement for synthesis in an inert environment and the final structure is solely determined by the bonding situation in the preceramic polymer.

Herein, NMR is used to determine the types of bonding present within the structure, FTIR to further confirm these bondings. Raman spectroscopy is employed for the characterization of the carbon phase and high-resolution TEM to determine the micro/nanostructure of the different SiOC samples. High temperature oxide melt solution calorimetry is used to determine the energetics and stabilities of the systems investigated. Such a combination of complementary techniques provides information as to how mixed bonding may contribute to the thermodynamic stability of SiOC PDCs and, in addition, offers indications about the role that hydrogen plays in this game.

## 2. Experimental Procedures

**Synthesis**: All chemicals were obtained and used as received from Sigma-Aldrich. Hydroquinone (HQ) ≥ 99%, tetramethylorthosilicate (TMOS), trimethoxyphenylsilane (TMOPS) 97%, and methanol were obtained from Sigma-Aldrich and used as received. The phenyl containing single-source non-mixed bonding precursors was prepared using 0.1, 4, and 8 wt% of HQ with respect to TMOS. The sol was prepared by adding water and TMOS to a mixture of methanol and HQ. The phenyl containing single-source mixed bonding precursor was prepared by adding water to a mixture of methanol and TMOPS. Ammonia was added dropwise to each reaction to obtain a pH of 9. The gelation process was done at 60 °C and the wet gel was further allowed to age in a 60 °C drying cabinet for 5 days. The resulting xerogels (HQ0.1_SG, HQ4_SG, HQ8_SG, PhSiO_1.5__SG) were dried in a 120 °C drying oven for 5 days. Pyrolysis of each polymer was performed both at 800 °C and 1100 °C under an argon atmosphere (100 °C/h heating rate, 2 h dwell time).

**Characterization**: Powder X-ray diffraction measurements were performed on a Bruker D2 Phaser diffractometer (accelerating voltage: 30 kV, emission current: 10 mA) operating in a Bragg–Brentano reflection geometry with a Cu Kα radiation source (λ = 1.54185 Å) and SSD160 1D detector. Data collection was conducted in the 2θ range of 5 to 120 degrees at a step size of 0.02. FTIR spectroscopy was run on a Varian 670 spectrometer. The powdered sample was pressed in a KBr pellet. Each spectrum consists of 32 scans and was recorded with a resolution of 4 cm^−1^. Raman spectroscopy was conducted using a Horiba HR800 micro-Raman spectrometer equipped with an Ar^+^ 514.5 nm laser. The measurements were performed with a grating of 600 g/mm and a confocal microscope at 50× g magnification with a 100 µm aperture. Neutral density filters attenuated the laser power on the samples in the range of 2 mW to 2 µW.

^29^Si magic angle spinning nuclear magnetic resonance (MAS NMR) and ^13^C(^1^H) cross-polarization (CP) MAS NMR spectra were collected using a 7 mm Bruker MAS probe and a Bruker Avance solid-state spectrometer operating at a Larmor frequency of 99.3 MHz for ^29^Si and 125.8 MHz for ^13^C. Powdered samples were packed in 7 mm ZrO_2_ rotors and spun at a rate of 7 kHz. ^29^Si single-pulse MAS spectra of all samples were collected using a 60° radio frequency pulse length of 2.6 µs and a 60 s recycle delay. Approximately 1500 free induction decays were collected and averaged for each ^29^Si MAS NMR spectrum. ^13^C(^1^H) CPMAS NMR spectra were collected using a 90° pulse length of 6.0 µs and a ^1^H spin-locking frequency of 38.5 kHz. A contact time of 1 ms and a recycle delay of 5 s was used. All spectra were collected under two pulse phase modulated proton decoupling [11,25] with a phase modulation angle of 15° and a pulse duration of 6.5 µs in the ^1^H channel (90° pulse length of 3.65 µs corresponding to an rf field strength of 68.5 kHz). Approximately 15,000 free induction decays were collected and averaged to obtain each ^13^C CPMAS NMR spectrum [11].

SiOC ceramic powders were prepared for high-resolution transmission electron microscopy via drop-coating. Thereby PDC powder is mixed with an appropriate solvent, here ethanol was used, and dispersed in an ultrasonic bath. Subsequently, one droplet of the suspension is placed onto a special TEM grid with a carrier carbon film (lacy carbon grid). To avoid charging under the incident electron beam, the samples were mildly coated with carbon. TEM/HRTEM investigations were carried out on a JEOL JEM 2100 F (Jeol, Akishima, Japan) equipped with a Schottky FEG with a nominal acceleration voltage of 200 kV. The EDS measurements were recorded with an Oxford XMAX 80 detector (Oxford Instruments plc, Tubney Woods, Abingdon, UK) attached to the microscope. The EELS data were acquired with the spectrometer attached to a JEOL JEM ARM 200F, equipped with a C_s_-corrected condenser system, with an acquisition time of 100 s, an energy dispersion of 0.3 eV per channel and an entrance aperture of 5 mm. The energy resolution measured at the zero-loss peak was 0.7–0.8 eV.

**Calorimetry**: The enthalpy of formation from crystalline components of the 800 °C and 1100 °C pyrolyzed ceramic samples, HQ0.1_SG, HQ4_SG, HQ8_SG, and PhSiO_1.5__SG was measured using high temperature oxide melt solution calorimetry. The sample (1–2 mg) was pressed into a pellet and dropped from room temperature into molten sodium molybdate (3Na_2_O•MoO_3_) solvent at 800 °C in a custom-built Tian-Calvet twin microcalorimeter in an oxidizing atmosphere. Oxygen gas was bubbled through the solvent at a rate of 5 mL/min to ensure the pellet reacted quickly and that oxidizing conditions were maintained. The gaseous products were flushed out of the system with an oxygen flow of ~90 mL/min. Eight pellets of each sample were dropped to obtain statistically reliable data. This methodology is well-established [18,20,26] and has been used previously on similar material systems [15,21,27].

## 3. Results and Discussion

### 3.1. Synthesis

The polymers synthesized as described above were pyrolyzed in an argon atmosphere at 800 °C and 1100 °C, resulting in two types of PDC ceramics, each with its distinct bonding character. While the PhSiO_1.5__SG material contained mixed bonding, the HQ4_SG was obtained without mixed bonding, although both polymers contained phenyl as a unique organic group and source of carbon. As shown in Figure 1a, the PhSiO_1.5__SG polymer precursor was synthesized to promote direct bonding between the silicon and the carbon of the phenyl group, which is maintained throughout pyrolysis. This polymer obtained is a polyhedral oligomeric silsesquioxane (POSS), as reported in previous studies utilizing a similar synthetic procedure [28]. In Figure 1b, it is illustrated that the silicon bonds to the phenyl through Si-O-C bridges, resulting in a SiOC ceramic without mixed bonding character. Herein, a new polysiloxane was formulated via the reaction of tetramethylorthosilicate with hydroxyquinone (HQ) to form a polysiloxane ceramic without any direct Si-C bonds in the structure.

Analysis of the full elemental composition was determined by Mikroanalytisches Labor Pascher in Germany and is tabulated in Table 1, along with the corresponding atomic percentages in parenthesis. The compositional analysis shows that a large amount of carbon was lost during pyrolysis of the HQ4_SG polymer, making impossible the direct comparison of two types of SiOC ceramics with similar compositions. Thus, the focus of this study lies in understanding the relationship between different chemical structures and final nano/micro-structures with or without mixed bonding characteristics. During the HQ4_SG polymer synthesis, it was noted that the color of the sample became brown during the drying process. Polysiloxanes and polysilsesquioxanes are generally white in color as was the case for PhSiO_1.5__SG. In air, hydroquinone can partially oxidize to benzoquinone, which actually can be of brown color. Since this synthesis was done under normal atmospheric conditions (air) in water, it is likely that a portion of the hydroquinone oxidized to benzoquinone. Benzoquinone, however, is not able to undergo hydrolysis and condensation reactions and, therefore, the added carbon most likely did not form bonds to the silicon backbone of the polymer, resulting in a significant loss of carbon during pyrolysis.

### 3.2. X-ray Powder Diffraction

The XRD patterns for all PhSiO_1.5__SG and HQ4_SG pyrolyzed samples are depicted in Figure 2. The amorphous halo found at 2 θ values of ~23° is caused by overlapping of amorphous silica and the carbon signal. The amorphous halo at ~6° seen in PhSiO_1.5__SG 800 °C can be attributed to the cage (cubane) structure formation of polysilsesquioxane [29]. The lack of crystalline SiC peaks indicates the ceramic remains amorphous at 1100 °C.

### 3.3. Fourier Transform Infrared Spectroscopy

The FTIR results are given in Figure 3 and Figure 4. First, the data of the polymers are presented, followed by the characterization of the pyrolyzed PDCs.

Polymers: The broad bands at ~3400 cm^−1^ assigned to O-H stretching [30] are seen in each polymer, most likely caused by O-H stretching related to Si–OH groups; the result of the hydration of SiO_4_ tetrahedra. This band is associated with hydrogen-bonded OH groups on the surface of SiO_2_ and physisorbed water [31]. Vibration bands between 960 and 990 cm^−1^ and at ~3650 cm^−1^ are also indicative of Si–OH group bending and asymmetric stretching, respectively [32,33,34,35]. The band at 3400 cm^−1^ is due to nearest neighbor Si-OH groups with strong hydrogen bonding interactions, while the band at 3650 cm^−1^ can be assigned to Si-OH groups that are partially shielded from similar neighboring groups and have weak interactions between the hydrogen of the Si-OH group and the oxygen in the Si–O–Si network [22]. The peak at 1027 cm^−1^ in the PhSiO_1.5__SG polymer is assigned to Si-O, a non-bridging oxygen [36], likely present due to incomplete reaction during the condensation process [37]. The bands between 3140 and 2890 cm^−1^ and the bands at 1594 and 1430 cm^−1^ in PhSiO_1.5__SG are characteristic of aromatic C–H stretching and phenyl C=C bonds [38,39] respectively, confirming that the phenyl R-groups in the polymer are present. Further, the peaks at 738 and 695 cm^−1^ confirm the presence of monosubstituted benzene [38], again confirming the aromatic substituent is a phenyl R-group. The fact that HQ4_SG polymer does not show any aromatic benzene bands could be due to the aromatic carbon groups starting to roll up into the pores of the polymer, which has been suggested by computations [22]. Bands between 1095 and 1225 cm^−1^ are assigned to Si-O-Si asymmetric vibrations [34,40,41] for both polymers. The strong band at 1095 cm^−1^ is also representative of Si–O–C stretching vibration [22]. This band is more pronounced in the HQ4_SG polymer as expected due to Si-O-C bridges used to link the phenyl groups to the polymer backbone.

In PhSiO_1.5__SG, the phenyl R-groups should only be bonded to the silicon in the polymer backbone and a band for Si–O–C stretching mode, if present, would be due to Si–O–CH_3_ end groups of the polymer. Further, HQ4_SG presents a shoulder band at ~1200 cm^−1^ due to Si-O-Si stretching. The bands between 790 and 795 cm^−1^, and bands between 460 and 495 cm^−1^, are assigned to Si–O–Si symmetric bending [22] and O–Si–O bending [36,41,42], respectively.

Pyrolyzed PDCs: FTIR revealed key changes that occurred during pyrolysis. In the PhSiO_1.5__SG spectra, the bands for the aromatic C–H stretch, phenyl C=C, and monosubstituted benzene, are no longer present after pyrolysis, indicating that the carbon in the sample transformed from a phenyl group to a sp^2^ carbon phase. Graphene-like carbons have poor IR transmittance [22] and the lack of bands in the IR spectra, indicating the presence of the “free” carbon phase, is not surprising. The Si–C band is still present at 800 °C at 875 cm^−1^ but is no longer seen in the FTIR spectra at 1100 °C. A band at 1365 cm^−1^ has been assigned to Si–CH_2_–Si bending [37,41,43]. This Si–C–Si linkage is evidence of the direct Si-C bonds that could indicate the presence of Si–C bonding at the interface between the silica and “free” carbon in a SiOC PDC with mixed bonding. The fact that this band is not observed after pyrolysis at 1100 °C is likely due to the loss of hydrogen at higher temperatures Bands between 1000–1225 cm^−1^ result from Si-O-Si asymmetric stretching vibrations and are seen in all pyrolyzed samples. This asymmetric stretching vibration band in PhSiO_1.5__SG occurs at ~1040 cm^−1^ for the sample pyrolyzed at 800 °C, and at ~1010 cm^−1^ for the sample pyrolyzed at 1100 °C, while in the HQ4_SG samples it occurs at ~1100 cm^−1^. The presence of framework defects results in shifts to lower values in FTIR spectra [22], which is directly related to the Si–O–Si bond angles. A comparison of the asymmetric stretching Si-O-Si mode for PhSiO_1.5__SG and HQ4_SG samples indicates that inserting carbon into the silica framework when mixed bonding is present, has the most impact on the band shift, as noted by the Si–O–Si asymmetric band in PhSiO_1.5__SG shifting to lower wavenumbers. When carbon is substituted into the silica framework, as in SiOCs with mixed bonding, the wavenumber of the associated Si–O–Si asymmetric stretching band is lowered as a result of the bond angle decrease [44].

The O–H stretching band present in both HQ4_SG samples, as well as the high hydrogen content seen in the compositional analysis, indicates that the structure is a composite of amorphous silica and sp^2^ carbon with terminal –OH groups on the silicon tetrahedra [45,46,47]. An intense band at 1095 cm^−1^ could also be caused by overlapping Si–O–Si/Si–O–C stretching modes and is present as the most intense band in both pyrolyzed HQ4_SG samples. This band could be indicative of covalent bonding at the interface between the silica and “free” carbon nanodomains or it could just be due to the Si–O–Si of the silica network.

### 3.4. Magic Angle Spinning Nuclear Magnetic Resonance

The ^29^Si MAS NMR of both PhSiO_1.5__SG samples, shown in Figure 5a,b, have peaks at ~−04, −72, −38, −11 and 7 ppm, which are characteristic peaks for SiO_4_, SiO_3_C, SiO_2_C_2_, SiC_4_ and SiOC_3_ respectively [11,48]. For the sample pyrolyzed at 800 °C, the SiO_4_ peak is at ~−102 ppm and at 1100 °C the peak is shifted to ~−107 ppm. One explanation for the downfield shift in the 800 °C sample is the presence of terminal hydroxyl groups on the silica. The ^29^Si chemical shift is very sensitive to the chemical environment of Si atoms in silicates and just one terminal -OH group on a silica tetrahedron will cause a chemical shift downfield. As seen in the FTIR for the PhSiO_1.5__SG samples, there is OH present in the 800 °C sample, but not in the 1100 °C sample. The compositional analysis also shows a decrease in the hydrogen content for the 1100 °C sample which is likely due to the loss of water during pyrolysis to the higher temperature. These results both support the conclusion that the downfield shift seen in the ^29^Si MAS NMR data is caused by terminal hydroxyl groups on the silica tetrahedra.

The percentage of each SiO_x_C_4−x_ mixed bond species, shown in Table 2, was calculated using the deconvoluted peaks from each ^29^Si MAS NMR spectrum. The PhSiO_1.5__SG sample pyrolyzed to 800 °C has a larger percentage of mixed bonding overall. A study by T. Liang et. al. [49] determined the effect of water in the pyrolysis environment. The pyrolysis of a polysiloxane in a mixture of argon and water vapor resulted in a SiOC with less mixed bonding than the same polysiloxane pyrolyzed in dry argon. PhSiO_1.5__SG has a relatively high hydrogen content, being higher at the lower pyrolysis temperature, which indicates that less water has been released. It is therefore likely that the water content evolved during pyrolysis at higher temperatures results in a lower percentage of mixed bonding in this sample.

The ^29^Si MAS NMR of HQ4_SG, shown in Figure 6a,b, indicates that the silicon bonding environment is different compared to the PhSiO_1.5__SG sample. The biggest difference is the absence of peaks that correspond to carbon substitution in the SiO_4_ tetrahedra in HQ4_SG. This confirms that there is very little or no mixed bonding in the HQ4_SG samples, but rather that this material is similar to a composite of silica with a minor fraction of sp^2^ “free” carbon. Consequently, the ^29^Si MAS NMR spectra for both HQ4_SG samples are composed of pure amorphous silica units with three isotropic peaks at ~−110, −99 and −91 ppm, corresponding to SiO_4_ (Q^4^), SiO_3_(OH) (Q^3^) and SiO_2_(OH)_2_(Q^2^) [50,51,52], respectively.

The percentage of each Q^n^ species, shown in Table 3, was calculated using the area under the curve of the deconvoluted peaks for each Si tetrahedron. The percentages of each Q^n^ species are consistent with sol-gel synthesis in basic conditions of silica, which tends to have a higher percentage of Q^4^ silica tetrahedra than materials synthesized in acidic conditions [53].

### 3.5. ^13^C(^1^H) Cross Polarization Magic Angle Spinning Nuclear Magnetic Resonance

The ^13^C CPMAS NMR spectra for the PhSiO_1.5__SG samples, given in Figure 7, have the characteristic isotropic chemical shift for sp^2^ carbon at ~127 ppm, with spinning side bands at ~242, 186, 74, 17 and −36 ppm, caused by the chemical shift anisotropy (CSA) [11]. ^13^C solid state MAS NMR peaks of aromatic carbons in various carbon materials ranges from 110 to 160 ppm, while peaks of aliphatic carbons range from 0 to 90 ppm [54,55]. In contrast, graphite and turbostratic carbon ^13^C NMR peaks range between ~120 and 130 ppm [56,57]. The isotropic peak in PhSiO_1.5__SG is therefore assigned to sp^2^ hybridized carbon in the form of graphite or turbostratic carbon. This carbon represents the “free” carbon phase of the SiOCH. Given the mixed bonding peaks seen in the ^29^Si MAS NMR spectra, it would be expected to see a peak from sp^3^ hybridized carbon in the range typical for silicon carbide of 10–25 ppm [58]. Although it is not obvious for either PhSiO_1.5__SG sample in the ^13^C NMR spectra that sp^3^ carbon is present, it is possible there is a small peak at ~18 ppm that overlaps with a spinning side band. The spectra of PhSiO_1.5__SG pyrolyzed at 800 °C were fitted with an sp^3^ peak at 18 ppm to demonstrate this aspect. In the simulated spectrum, adding the sp^3^ carbon peak results in a peak at 18 ppm that fits more closely to the observed intensity, indicating that it is quite possible that this peak is present. Due to the background noise of the spectrum of the 1100 °C sample, it is more difficult to determine whether an overlapping peak is present and therefore a simulation with an additional sp^3^ peak was omitted.

The spectra for HQ4_SG samples are shown in Figure 8. These spectra are not very well resolved due to possible overlapping carbon signals. Each of the spectra were fitted with three isotropic peaks at ~125, 104 and 26 ppm, assigned to sp^2^ graphitic carbon, carbon nanotubes (CNTs)/carbon nanoscrolls (CNSs) and sp^3^ carbon, respectively. The graphitic carbon peak in each spectrum was also assigned here, similar to the PhSiO_1.5__SG sample. The CNT/CNS peak can explain the overall broadening of the spectra. In general, peaks in ^13^C MAS NMR spectra of single walled CNTs are broad due to the inhomogeneous dispersion of chemical shifts caused by differing nanotube chiralities, lengths and adjacent defects [59,60,61,62]. Multi walled CNTs on the other hand show broadening in the isotropic peak mainly due to diamagnetic shielding [63,64] caused by CNTs enclosed by other CNTs [65] The graphitic sp^2^ carbon peak is thought to simply originate from an aromatic carbon network that has not fully reacted to form rolled-up structures. The third possible peak seen in the ^13^C(^1^H) MAS NMR spectra of HQ4_SG is at ~26 ppm. This peak is in the range of aliphatic carbon [54] underlining the presence of carbon nanoscrolls with saturated edges instead of CNTs. Note that no evidence of Si-C bonding in these samples was found in the corresponding ^29^Si MAS NMR spectra.

### 3.6. Micro-Raman Spectroscopy

The Raman spectra of all PhSiO_1.5__SG and HQ4_SG samples, depicted in Figure 9, have extensive noise introduced due to fluorescence in the samples when using a 514.5 nm laser source. Despite this, clear D and G bands are seen at ~1330 cm^−1^ and 1600 cm^−1^, which is consistent with sp^2^ carbon, where the characteristic Raman bands are the disordered carbon D band at ~1350 cm^−1^ and the in-plane bond stretching G band at ~1590 cm^−1^, as also seen in the fitted Raman spectra in Figure 9. Additional bands at ~105 cm^−1^ and 150 cm^−1^ are assigned to radial like breathing modes (RBLMs). These bands are similar to those reported for CNTs as low energy active Raman modes that result in bands due to the radial breathing modes. These radial breathing modes are a result of the synchronous expanding and contracting of the nanotube in the radial direction [66,67]. Thus, the RBM position is related to the diameter of the CNT and is not seen in other types of sp^2^ carbon [68]. When the carbon phase is rolled up in the pores of silica, it can form carbon nanoscrolls that have more than one RLBM band, as opposed to SWCNTs, which are expected to have only one RBM [22]. The RLBM bands of HQ4_SG samples are shifted slightly to lower wavenumber, which indicates different diameters of the nanoscrolls [69]. After 200 cm^−1^, the overtone of the fluorescence is too strong, and no other bands are seen. From the Raman data, it is concluded that spiral-like structures with rolled-up geometry and open edges are present in these samples, in particular in the HQ4_SG 800 °C sample, however, further analysis is necessary to determine their specific nature.

The fitting of D and G bands offers the possibility to evaluate the carbon-cluster size L_a_ by using the formula reported by Cançado (Equation (1)) [70].
(1)$$L_{a\;}\;\left(\mathrm{nm}\right)=\frac{560}{E_{l}^{4}}{\left[\frac{A_{D}}{A_{G}}\right]}^{-1}$$
with *L_a_* being the size of the carbon domains along the six-fold ring plane (lateral size between line defects), *E_l_* the energy of the laser used in the study (2.41 eV corresponding to the laser wavelength of 514.5 nm) and *A_D_*/*A_G_* the intensity ratio of the *D* and *G* modes.

Lorentzian and Gaussian curve fitting of the Raman bands was performed in order to extract the *A_D_*/*A_G_* intensity ratios and to determine the size of the free carbon clusters formed in the PDC ceramics. The peak fitting was done including the minor *D*′′ and *A* bands (see inset in Figure 9a). There is a strong dependency of the *A_D_*/*A_G_* ratio on the degree of disorder in graphene-like materials [71]. The disorder is quantified as a function of point-like defects, precisely on the inter-defect distance, *L_D_*, as calculated with Equation (2). As reference, for pristine graphene *L_D_*→∞ and for fully disordered graphene *L_D_*→0.
(2)$$L_{D}\;\left(\mathrm{nm}\right)=\sqrt{\frac{4300}{E_{L}^{4}\left(eV^{4}\right)}{\left[\frac{A_{D}}{A_{G}}\right]}^{-1}}$$

The inter-defect distances *L_D_*, the lateral cluster sizes *L_a_* and the ratio of *D* and *G* are summarized in Table 4.

As observed in Table 4, there is a blue shift of the D bands of HQ4_SG samples at both 800 and 1100 °C as compared with the PhSiO_1.5__SG ceramics. The blue shift originates from compression and a quantum effect of the carbon lattice into the silica porous matrix. This observation underlines the fundamental structural difference in-between the carbon phase of the SiOC ceramic samples with and without mixed bonds in their structure. The large compressive stress affecting the C=C bonds in a highly defective carbon phase can be observed also by the blue shift of the G band at ~1600 cm^−1^, as compared to other carbonaceous materials.

A relatively high *A_D_*/*A_G_* ratio (2.3 to 5.5 as a sign of disorder) was registered for all samples, except for HQ4_SG 800. The slight disorder of the samples is also underlined by the presence of the D″ band (shoulder at ~1150 −1200 cm^−1^ being attributed to the presence of sp^2^–sp^3^ C–C and C=C bonds) and of the A band (at ~1500 cm^−1^, corresponding to the fraction of amorphous carbon contained in the samples). The disorder originates from the presence of sp^3^ hybridized C edges of the graphene scrolled layers and of the pores, as shown also by the MAS NMR study. The lateral size *L_a_* (between line defects) follows the same trend as the *A_D_*/*A_G_* ratio with the lowest value (highest disorder) for the sample PhSiO_1.5__SG_1100 °C (6.8 nm) and the highest value (highest order) for the HQ4_SG 800 °C sample (44.2 nm). For both types of SiOC ceramics, with and without mixed bonds, the structural rearrangements seem to increase disorder at 1100 °C as compared to 800 °C. For the PhSiO_1.5__SG samples, the release of hydrogen from 800 to 1100 °C is probably a key factor for the increase in the disorder with increasing temperature. The observed trend for the lateral carbon size *L_a_* was followed by the inter-defect distance *L_D_* (point-like defects), providing the lowest defect density for the sample HQ4_SG 800 °C and the highest defect density for PhSiO_1.5__SG 1100 °C.

### 3.7. High-Resolution Transmission Electron Microscopy

To investigate the local microstructural evolution of the SiOC ceramics with or without mixed bonding structure and, in addition, to gain information on the nature and organization of possible nanocarbon phases present in both materials, transmission electron microscopy (TEM) in conjunction with energy-dispersive X-ray spectroscopy (EDS) and electron energy loss spectroscopy (EELS) were performed. Here, both samples synthesized at the lower pyrolysis temperature of 800 °C were compared, in order to analyze the early stage of polymer-to-ceramic transition. As shown in Figure 10, both samples are completely amorphous, as can be seen from the diffuse halos in the FFT/SAED images (see insets in (a) and (c)), being consistent with the XRD-data presented in Figure 2. In addition, the inverse Fast-Fourier-Transform (Inv-FFT) images given in the insets of Figure 10b,d, also reveal no indication of local ordering or the initiation of local nucleation. This finding is in general, not unexpected, considering the relatively low pyrolysis temperature of 800 °C. However, a rather unexpected result of the HRTEM imaging is that no indication of the presence of any carbon phase, even not for the PhSiO1.5_SG 800 °C sample, could be observed. As shown in Table 1, the chemical analysis of PhSiO1.5_SG 800 °C gave a carbon content of ca. 44 wt.%, which was expected to be “seen”. On the other hand, considering a high volume fraction of highly disordered or even amorphous carbon being present in this sample, a distinction from the at such pyrolysis temperature typically amorphous SiOC matrix phase will not be possible by HRTEM, in particular, since basically only light elements, with the exception of Si compose both materials.

One interesting aspect that can be noted when comparing the bright-field (BF) images shown in Figure 10a,b versus Figure 10c,d is that the HQ4_SG 800 °C sample is highly porous (compare also Figure 11), while the PhSiO_1.5__SG 800 °C material is fully dense without any local porosity or local contrast variation that could be imaged. Even on a 2 nm scale, as given in the inset in (d), no porosity was detected, while in (b) the observed pore size ranges between 5–10 nm in diameter. Therefore, the HQ4_SG 800 °C sample is characterized by a highly mesoporous fully amorphous micro/nanostructure. It is assumed that the small pores are accommodated within a matrix of rounded silica particles being of similar size, ranging approximately between 5–10 nm in diameter. Considering also the chemical EDS analysis, it is fair to state that this sample is predominantly composed of nanosized silica spheres, with the exception of about 3 wt.% of the carbon phase (see Table 1). It was expected that the carbon phase is mainly present along the pore walls within the matrix, as reported earlier by Kroll et al. [72] based on density functional theory (DFT) calculations. However, such a lining of the pore walls was not detected, possibly due to (i) the rather low volume fraction of carbon within this sample (as compared to the large intrinsic pore volume) and (ii) that at the low temperature of 800 °C the carbon phase is assumed to still be rather disordered.

The PhSiO_1.5__SG 800 °C sample, on the other hand, contains a comparably high fraction of carbon with ca. 44 wt.%, which however also could not be depicted via HRTEM. One potential reason for not detecting a separate carbon phase would be that the intrinsic polymer network structure is still intact and no phase separation has yet occurred. The rather high content of residual hydrogen of ~~19 at.% may be indicative of this assumption. On the other hand, considering that this sample is fully dense and, moreover, that the carbon phase forms a percolation network with an individual structural size of approximately 2–5 nm, the question arises why such a phase was not detectable by HRTEM (with a point resolution of ca. 0.2 nm)? Here it becomes obvious that, although TEM is commonly considered a very powerful technique, it also has its limitations. Consider a typical TEM sample close to the materials’ edge being on the order of 20–50 nm in thickness and a percolating, disordered phase within the amorphous matrix of only one-tenth of the slab size. Since, unfortunately, we always have to deal with the projection of such microstructural arrangements/features within this sample slab onto the imaging screen, this percolation phase will consequently not be visible in the corresponding (projected) image; compare also both Inv-FFT insets in Figure 11. The fact that we are dealing here with more or less disordered structures embedded in a similarly disordered, namely amorphous SiOC matrix, prevents the detection of the carbon phase.

Lastly, it should be noted that EELS analysis (not shown here) of the low-temperature pyrolysis samples was unfortunately not successful. Both samples reacted rather sensitive under a focused small electron beam and tended to locally decompose during analysis. This is assumed to be a consequence of the rather high residual hydrogen content of ca. 5 versus 20 at.% in the HQ4_SG 800 °C and the PhSiO_1.5__SG 800 °C sample, respectively, whereby local rearrangements of the SiOC network structure and local hydrogen release may have been initiated via the incident electron beam.

## 4. Thermochemistry

High temperature oxidative drop solution calorimetry was used to determine the enthalpy of oxidation and the enthalpies of formation from both elements and components (SiO_2_, SiC, C, H_2_O) of the SiOC(H) PDCs. This method has been well developed and extensively used for the thermodynamic investigation of PDCs [12,13,14,15,21,22,73]. The calorimetric experiment directly measured the enthalpy of drop solution (ΔH_ds_) of the reaction, which can be calculated per mole based on the sample composition. Here one mole of sample is defined as Avogadro’s number of atoms (Si + O + C + H), i.e., a gram atom. Due to the high hydrogen content of these samples and the excess oxygen present, relative to that needed to form SiO_2_, it is assumed that the hydrogen is all bound in water. The high temperature oxidation of the sample occurs as shown in Equation (3).
(3)$${\mathrm{Si}}_{a}O_{\text{b-d}/2}C_{c}\;\frac{d}{2}\;H_{2}O\;(\mathrm{solid},\;25\;{}^{\circ}C)+\left(a+c-\frac{b}{2}+\frac{d}{4}\right)\;O_{2}\;(\mathrm{gas},\;802\;{}^{\circ}C)\text{→}\;a\;{\mathrm{SiO}}_{2}\;(\mathrm{cristobalite},\;802\;{}^{\circ}C)+c\;{\mathrm{CO}}_{2}\;(\mathrm{gas},\;802\;{}^{\circ}C)+\frac{d}{2}\;H_{2}O\;(\mathrm{gas},\;802\;{}^{\circ}C)$$

The enthalpy of oxidation at room temperature and the enthalpy of formation from both elements and components at room temperature were then calculated using thermodynamic cycles shown in Table 5.

All experimental and calculated enthalpies can be found in Table 6. The oxidation at room temperature for all samples is exothermic with PhSiO_1.5__SG samples being more exothermic than HQ4_SG samples. The enthalpy of oxidation is directly related to the amount of carbon present in the PDC. The enthalpy of formation from elements of all samples is exothermic with HQ4_SG samples being more exothermic. Since these values are exothermic, the samples are thermodynamically stable when compared to the elements. The enthalpy of formation from components, SiO_2_ (cristobalite), SiC, and C (graphite) for HQ4_SG pyrolyzed at 800 °C is endothermic with a value of 20.0 ± 0.9 kJ/g⋅at. This positive enthalpy indicates that HQ4_SG800 is energetically unstable with respect to the crystalline components. The instability may be due to a destabilizing effect caused by the presence of carbon nanoscrolls which may have, as their analogs CNTs, a positive enthalpy of formation from graphite [22,74,75]. The other three samples, HQ4_SG1100, PhSiO_1.5__SG800 and PhSiO_1.5__SG1100 all have negative enthalpies of formation, values ranging from ~−3 to −14 kJ/g⋅at, indicating that they are energetically stable when compared to their crystalline components. The SiOC(H) PDCs with mixed bonding are more energetically stable than their non-mixed bonding counterparts pyrolyzed at the same temperature. While it has been suggested that mixed bonding is the main factor responsible for the thermodynamic stabilities of SiOC PDCs, there must be other factors that contribute to thermodynamic stability, since HQ4_SG1100, with no mixed bonding, is also energetically stable. However, the lack of success in controlling composition makes it impossible to draw definitive conclusions about the relative importance of composition and microstructure.

## 5. Conclusions

The synthesis of two different preceramic polymers with phenyl R groups, one in which the phenyl group is directly bonded to silicon through Si-C bonds and one with the phenyl group bonded to the silicon through Si-O-C bridges, result in SiOC PDCs with significantly different bonding, particularly at the interface between the silica rich nanodomains and the sp^2^ “free” carbon phase/nanodomains. The synthesis route followed for the PhSiO_1.5__SG material, gave direct silicon-carbon bonds in the polymer, resulting in a SiOC PDC with mixed bonding at the interface between the silica-rich and carbon-rich regions. The absence of direct Si-C bonds in the polymer network generates a SiOC PDC with no mixed bonding at the interface, as is the case for the HQ4_SG material. These nanodomain bonding patterns are assumed to be relevant for nearly all of the “free” sp^2^ carbon nanodomains in both PDCs, which is confirmed by the presence of radial breathing-like mode bands in the Raman spectra. The specific nature of these curled carbon nanoregions is however not well known and further studies are needed to confirm their morphology. The 2–5 nm in size mesopores found by TEM in the silica-rich structure of HQ4_SG samples accommodate the spiral-like rolled-up graphene sheets with open sp^3^ saturated edges. The energetics, as determined by high temperature oxide melt solution calorimetry, confirms that the SiOC PDC materials with mixed bonding are thermodynamically more stable than those without. It became apparent that there are other factors contributing to the thermodynamic stability besides mixed bonding alone in SiOC PDCs, as for example the atomistic structure of the “free” carbon nanodomains.

## Figures and Tables

**Figure 1 materials-14-04075-f001:**
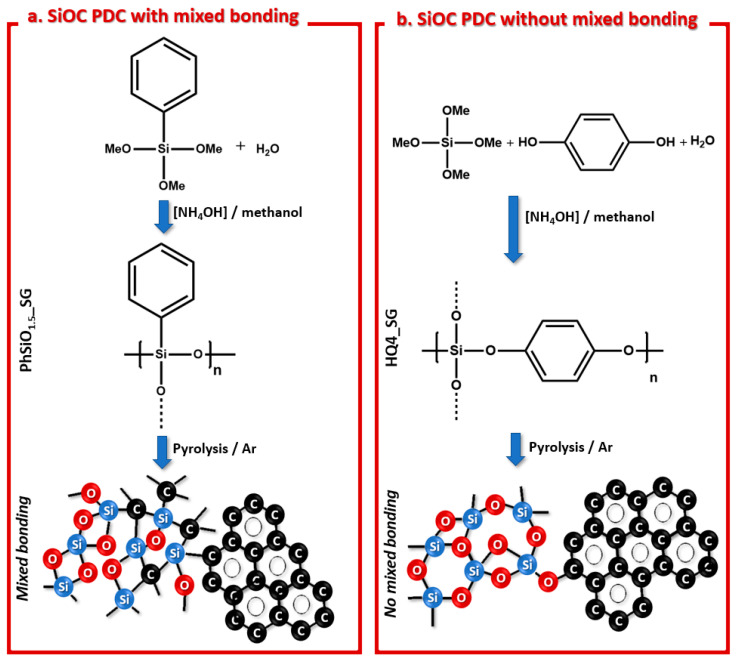
(**a**) Synthesis of PhSiO_1.5__SG using trimethoxyphenylsilane (TMOPS) precursor to synthesize a polymer with the phenyl group bonding directly to silicon resulting in a SiOC PDC with mixed bonding. (**b**) Synthesis of HQ4_SG using hydroquinine (HQ) and tetramethylorthosilicate (TMOS) precursors to synthesize a polymer with the phenyl group bonding to the silicon through Si-O-C bridges resulting in a SiOC PDC without mixed bonding.

**Figure 2 materials-14-04075-f002:**
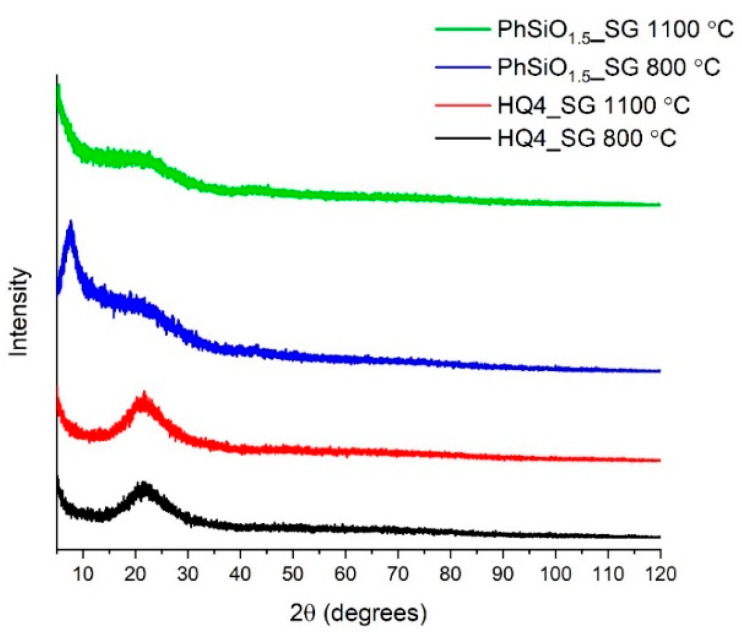
XRD patterns of PhSiO_1.5__SG and HQ4_SG pyrolyzed to 800 °C and 1100 °C.

**Figure 3 materials-14-04075-f003:**
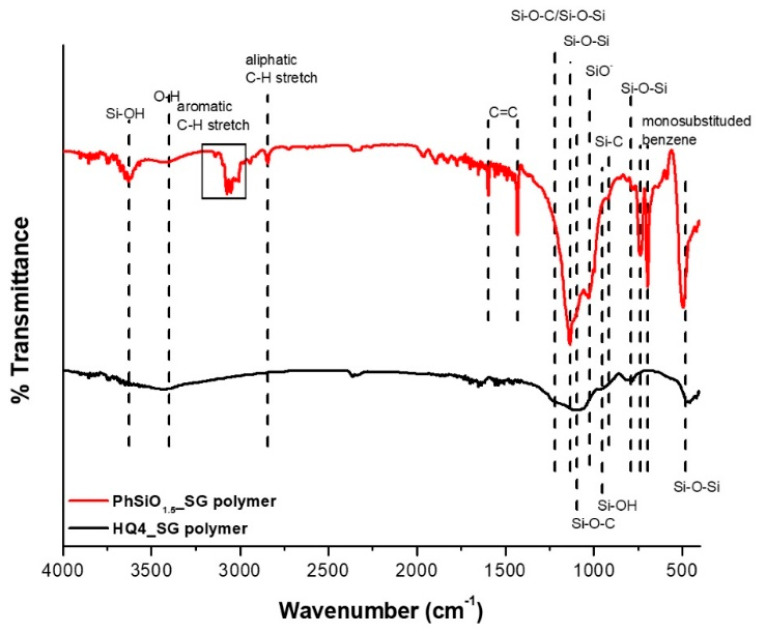
FTIR spectra of HQ4_SG and PhSiO_1.5__SG polymers.

**Figure 4 materials-14-04075-f004:**
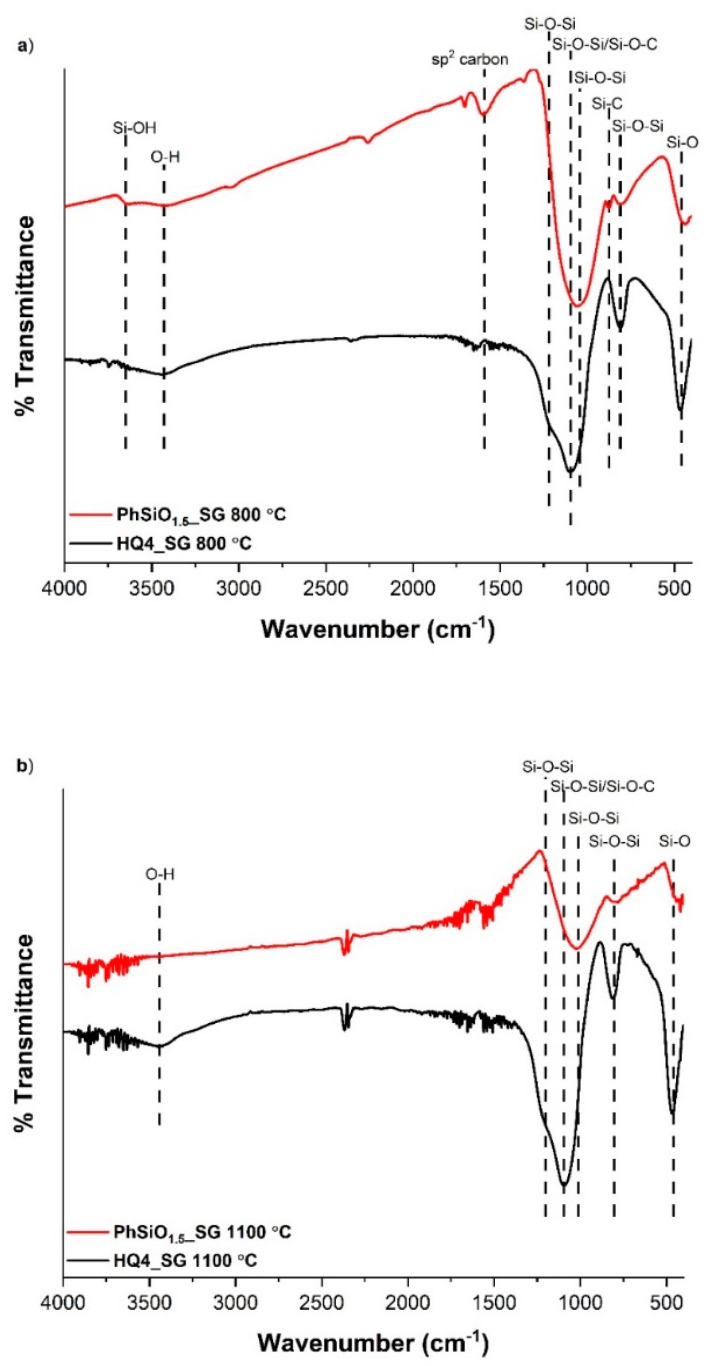
FTIR spectra of HQ4_SG and PhSiO_1.5_SG pyrolyzed to (**a**) 800 °C and (**b**) 1100 °C.

**Figure 5 materials-14-04075-f005:**
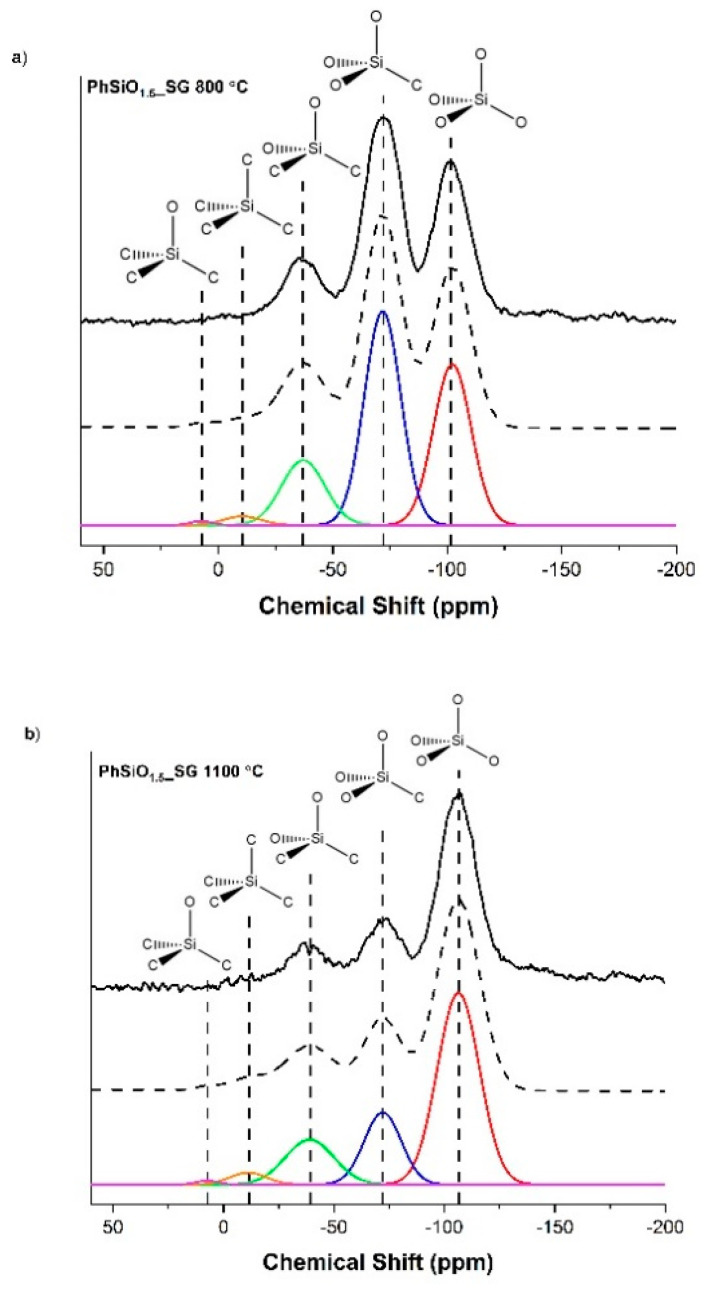
^29^Si MAS NMR for PhSiO_1.5__SG samples. (**a**) Sample pyrolyzed to 800 °C with peaks at ~−102 ppm, −72 ppm, −38 ppm, −11 ppm and 7 ppm, corresponding to SiO_4_, SiO_3_C, and SiO_2_C_2_, SiC_4_ and SiOC_3_ tetrahedra, respectively. (**b**) Sample pyrolyzed to 1100 °C with peaks at ~−107 ppm, −72 ppm, −38 ppm, −11 ppm and 7 ppm corresponding to the same structural units given in (**a**).

**Figure 6 materials-14-04075-f006:**
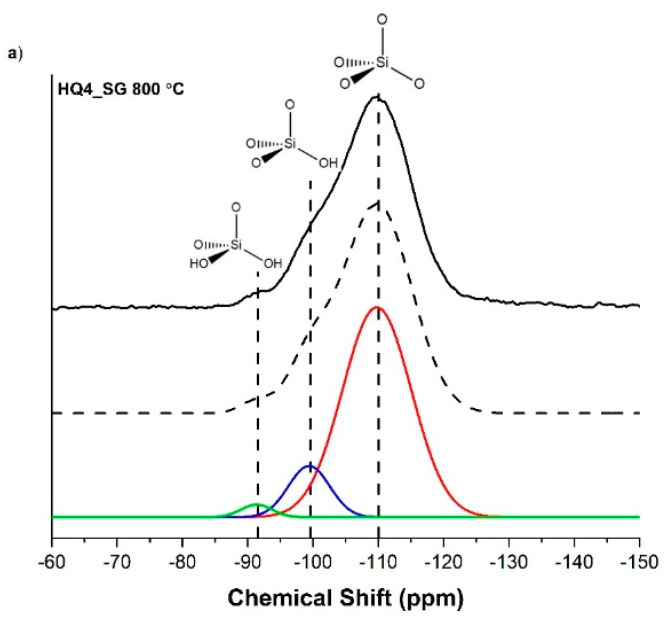
^29^Si MAS NMR for HQ4_SG samples with peaks at ~−110 ppm, −100 ppm and −92 ppm, corresponding to SiO_4_, SiO_3_(OH) and SiO_2_(OH)_2_ tetrahedra, respectively; for (**a**) the sample pyrolyzed to 800 °C and (**b**) the sample pyrolyzed to 1100 °C.

**Figure 7 materials-14-04075-f007:**
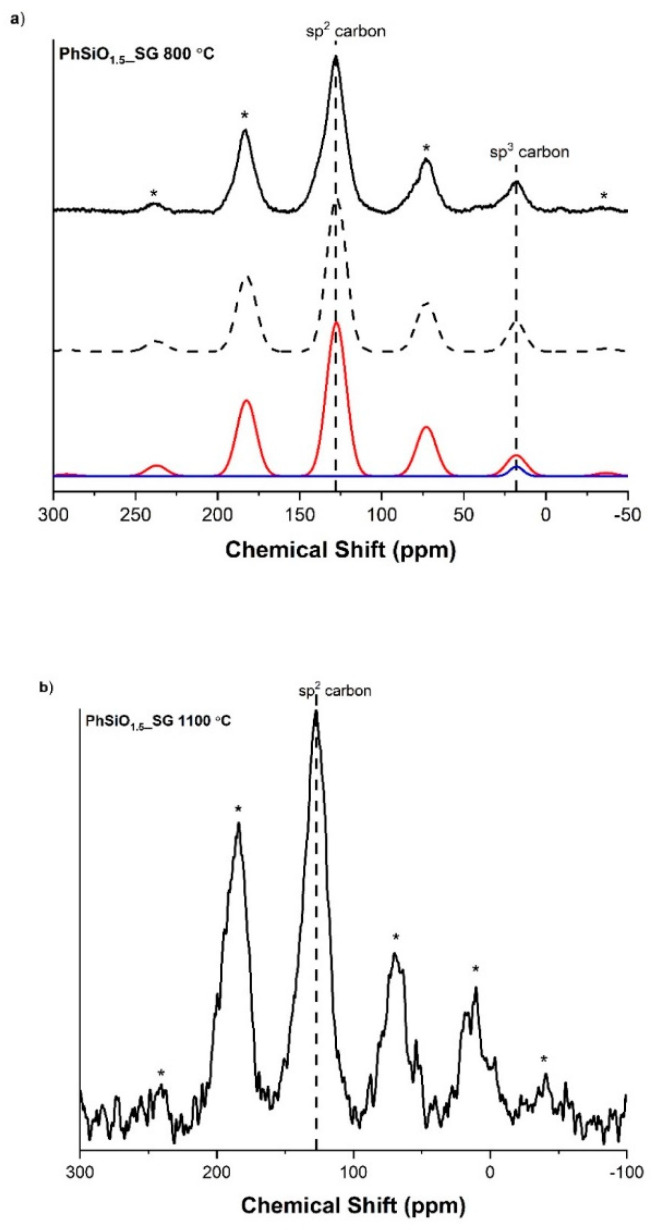
^13^C(^1^H) CPMAS NMR for PhSiO_1.5__SG samples. (**a**) Sample pyrolyzed to 800 °C with peaks at ~127 ppm and 18 ppm, corresponding to sp^2^ hybridized carbon and sp^3^ hybridized carbon, respectively. (**b**) Sample pyrolyzed to 1100 °C with an isotropic peak at ~127 ppm, corresponding to sp^2^ hybridized carbon. The stars * denote spinning side bands.

**Figure 8 materials-14-04075-f008:**
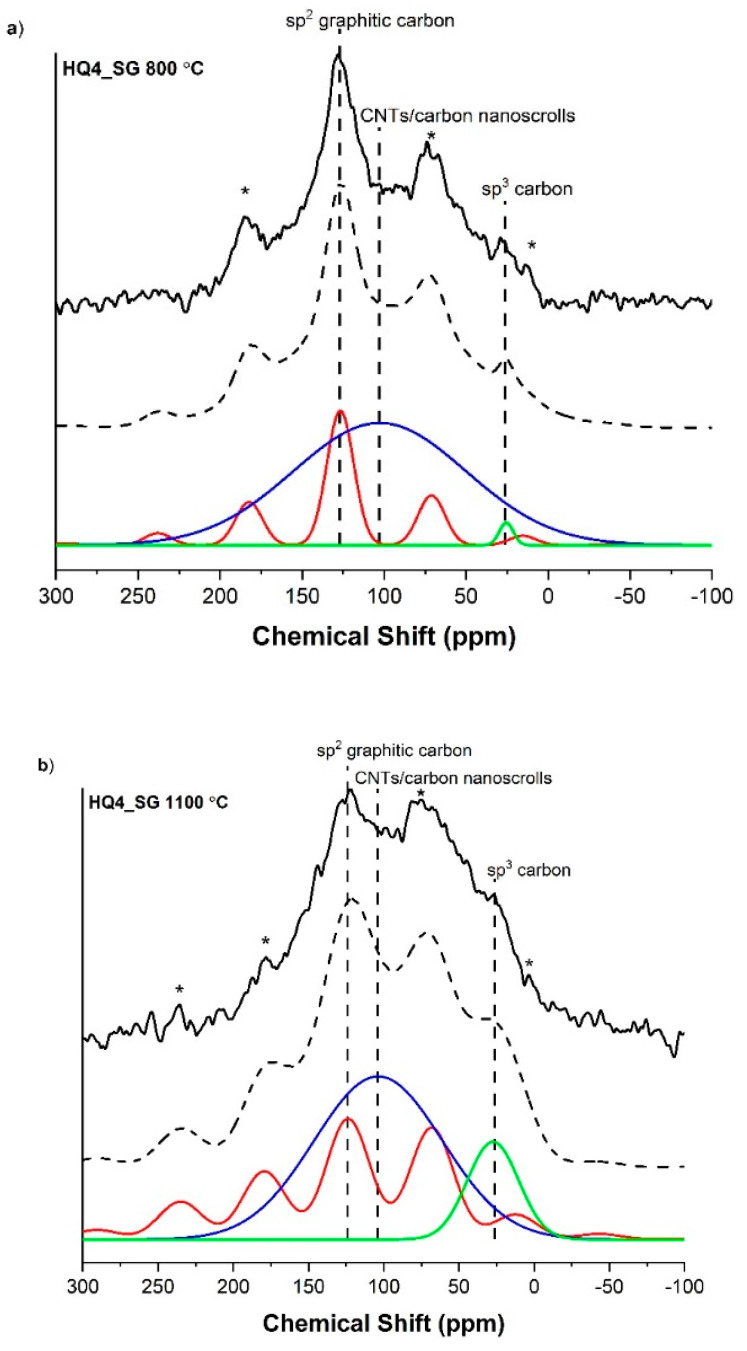
^13^C(^1^H) CPMAS NMR of the HQ4_SG samples with isotropic peaks at ~127 ppm, 103 ppm and 26 ppm, corresponding to sp^2^ hybridized graphitic carbon, CNT/carbon nanoscrolls and sp^3^ hybridized carbon, respectively; for (**a**) the sample pyrolyzed to 800 °C and (**b**) the sample pyrolyzed at 1100 °C. The stars * denote spinning side bands.

**Figure 9 materials-14-04075-f009:**
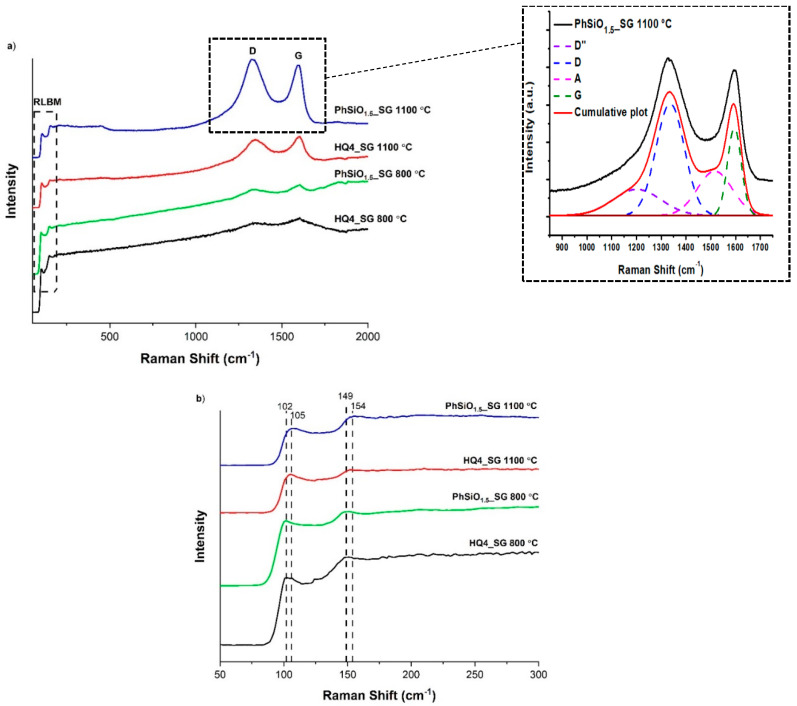
Raman spectra of PhSiO_1.5__SG and HQ4_SG samples. (**a**) The entire spectra from 50 to 2000 cm^−1^ with inset showing the Raman fitting of PhSiO_1.5__SG 1100 °C and (**b**) the spectra from 50 to 300 cm^−1^ to show details of the RLBM bands.

**Figure 10 materials-14-04075-f010:**
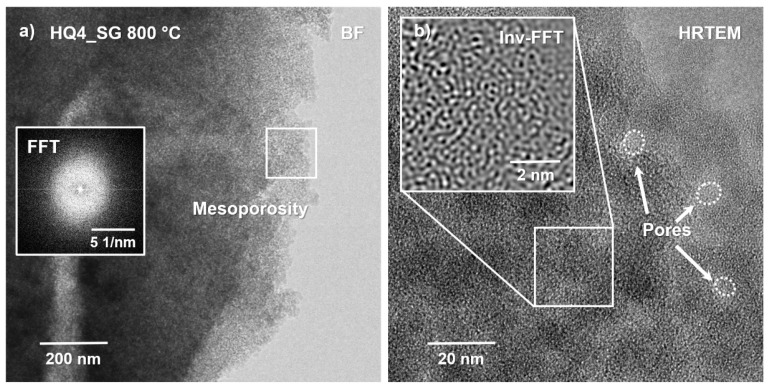
Bright-field (BF) and high-resolution transmission electron microscopy (HRTEM) images of (**a**,**b**) the HQ4_SG 800 °C sample, while in (**c**,**d**) the PhSiO_1.5__SG 800 °C material is depicted. Both samples are fully amorphous, as can be seen from the FFT/SAED insets in (**a**,**c**) as well as from the inverse Fast-Fourier-Transform images (Inv-FFT) given in (**b**,**d**).

**Figure 11 materials-14-04075-f011:**
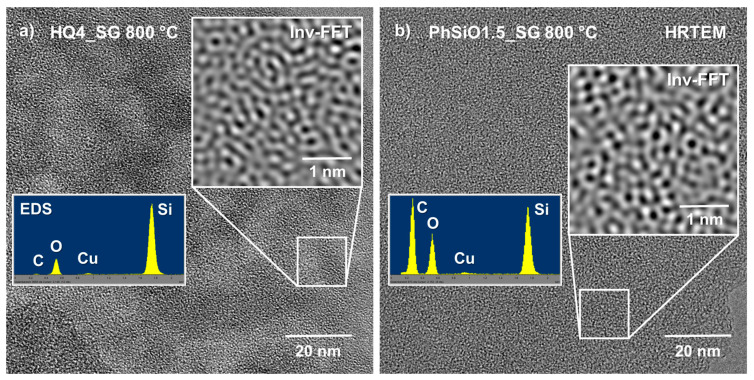
High-resolution transmission electron microscopy images comparing the two materials studied (**a**) HQ4_SG 800 °C and (**b**) PhSiO_1.5__SG 800 °C. The Inv-FFT insets again underline the fully amorphous nature of both samples. However, a major difference found in these samples is the respective chemical composition with a much higher carbon content found in the PhSiO_1.5__SG 800 °C sample (see EDS insets), as compared to the mainly of Si and O composed material HQ4_SG 800 °C. Note that the EDS results are consistent with the chemical analysis reported in Table 1.

**Table 1 materials-14-04075-t001:** Analyzed composition of SiOCH ceramic recorded as wt% (at.%).

Sample	Si (wt.% (at.%))	O (wt.% (at.%))	C (wt.% (at.%))	H (wt.% (at.%))
HQ4_SG 800 °C	43.3 (28.6)	53.4 (61.9)	3.05 (4.71)	0.26 (4.8)
HQ4_SG 1100 °C	42.7 (28.3)	50.5 (58.8)	5.02 (7.78)	0.28 (5.2)
PhSiO_1.5__SG 800 °C	24.8 (11.2)	28.6 (22.7)	44.36 (46.98)	1.51 (19.1)
PhSiO_1.5__SG 1100 °C	27.2 (14.2)	25.1 (23.0)	46.51 (56.81)	0.41 (6.0)

**Table 2 materials-14-04075-t002:** Percentage of each SiO_x_C_4−x_ Si species calculated from ^29^Si MAS NMR.

Sample	Si Species (%)				
	SiO_4_	SiO_3_C	SiO_2_C_2_	SiOC_4_	SiC_4_
PhSiO_1.5__SG 800 °C	35.55	45.57	16.44	0.51	1.93
PhSiO_1.5__SG 1100 °C	60.20	19.77	16.23	0.61	3.19

**Table 3 materials-14-04075-t003:** Percentage of each Q^n^ Si species calculated from ^29^Si MAS NMR.

Sample	Q^n^ Si Species (%)		
	SiO_4_ (Q^4^)	SiO_3_(OH) (Q^3^)	SiO_2_(OH)_2_ (Q^2^)
HQ4_SG 800	85.2	12.5	2.3
HQ4_SG 1100	80.6	17.2	2.2

**Table 4 materials-14-04075-t004:** Raman graphitization parameters for all samples reported in this study.

Sample	*ω*_D_ (cm^−1^)	*ω*_G_ (cm^−1^)	*A_D_*/*A_G_*	*L_a_* (nm)	*L_D_* (nm)	ο (1/nm^2^)
HQ4_SG 800 °C	1345	1599	0.9	44.2	18.4	0.0029
HQ4_SG 1100 °C	1350	1601	2.9	13.1	10.0	0.0099
PhSiO_1.5__SG 800 °C	1335	1595	2.3	16.2	11.1	0.0080
PhSiO_1.5__SG 1100 °C	1333	1597	5.5	6.8	7.2	0.0190

*ω* represents the exact wavenumber position of the respective Raman band.

**Table 5 materials-14-04075-t005:** Thermodynamic cycles for SiOC(H) heat of formation.

Reaction	Enthalpy (∆H)
Enthalpy of oxidation (∆H^0^_ox_) at 25 °C	-
(1) Si_a_O_b-d/2_C_c_ + $\frac{d}{2}$ H_2_O (solid, 25 °C) + $\left(a+c-\frac{b}{2}+\frac{d}{4}\right)$ O_2_ (gas, 802 °C)→a SiO_2_ (cristobalite, 802 °C) + c CO_2_ (gas, 802 °C) + $\frac{d}{2}\;$ H_2_O (gas, 802 °C)	∆H_1_ = ∆H_ds_ (kJ/g at.)
(2) SiO_2_ (cristobalite, 25 °C)→SiO_2_ (cristobalite, 802 °C)	∆H_2_ = 50.1 kJ/mol
(3) H_2_O (liquid, 25 °C)→H_2_O (gas, 802 °C)	∆H_3_ = 73.1 kJ/mol
(4) O_2_ (gas, 25 °C)→O_2_ (gas, 802 °C)	∆H_4_ = 25.3 kJ/mol
(5) CO_2_ (gas, 25 °C)→CO_2_ (gas, 802 °C)	∆H_5_ = 37.5 kJ/mol
Si_a_O_b-d/2_C_c_ + $\frac{d}{2}$ H_2_O (solid, 25 °C) + $\left(a+c-\frac{b}{2}+\frac{d}{4}\right)$ O_2_ (gas, 25 °C)→a SiO_2_(cristobalite, 25 °C) +c CO_2_ (gas, 25 °C) + $\frac{d}{2}$ H_2_O (liquid, 25 °C)	∆H^0^_ox_ (kJ/g at.) = ∆H_1_ − a ∆H_2_ − $\frac{d}{2}$ ∆H_3_ + $\left(a+c-\frac{b}{2}+\frac{d}{4}\right)$ ∆H_4_ − c ∆H_5_
Enthalpy of formation from the elements ∆H^0^_f,elem_ at 25 °C	-
(1) Si_a_O_b-d/2_C_c_ + $\frac{d}{2}$ H_2_O (solid, 25 °C) + $\left(a+c-\frac{b}{2}+\frac{d}{4}\right)$ O_2_ (gas, 25 °C)→a SiO_2_(cristobalite, 25 °C) +c CO_2_ (gas, 25 °C) + $\frac{d}{2}\;$ H_2_O (liquid, 25 °C)	∆H_1_ = ∆H^0^_ox_ (kJ/g·at.)
(2) Si (solid, 25 °C) + O_2_ (gas, 25 °C)→SiO_2_ (cristobalite, 25 °C)	∆H_2_ = −908.4 ± 2.1 kJ/mol
(3) H_2_ (gas, 25 °C) + 1/2 O_2_ (gas, 25 °C)→H_2_O (liquid, 25 °C)	∆H_3_ = −285.8 ± 0.1 kJ/mol
(4) C (solid, 25 °C) + O_2_ (gas, 25 °C)→CO_2_ (gas, 25 °C)	∆H_4_ = −393.5 ± 0.1 kJ/mol
a Si (solid, 25 °C) + $\frac{b}{2}$ O_2_ (gas, 25 °C + c C (solid, 25 °C) + $\frac{d}{2}$ H_2_ (gas, 25 °C)→Si_a_O_b-d/2_C_c_ + $\frac{d}{2}$ H_2_O (solid, 25 °C)	∆H^0^_f,elem_ (kJ/g-at.) = −∆H_1_ + a ∆H_2_ + $\left(\frac{d}{2}\right)$ ∆H_3_ + c ∆H_4_
Enthalpy of formation from the components ∆H^0^_f,comp_ at 25 °C	
(1) a Si (solid, 25 °C) + $\frac{b}{2}$ O_2_ (gas, 25 °C) + c C (solid, 25 °C) + $\frac{d}{2}\;$ H_2_ (gas, 25 °C)→Si_a_O_b-d/2_C_c_ + $\frac{d}{2}\;$ H_2_O (solid, 25 °C)	∆H_1_ = ∆H^0^_f,elem_ (kJ/g·at.)
(2) Si (solid, 25 °C) + C (solid, 25 °C)→SiC (solid, 25 °C)	∆H_2_ = −73.2 ± 6.3 kJ/mol
(3) Si (solid, 25 °C) + O_2_ (gas, 25 °C)→SiO_2_ (cristobalite, 25 °C)	∆H_3_ = −908.4 ± 2.1 kJ/mol
(4) H_2_(gas, 25 °C) + O_2_ (gas, 25 °C)→H_2_O (liquid, 25 °C)	∆H_4_ = −285.83 ± 0.04 kJ/mol
$\left(a-\frac{b}{2}+\frac{d}{4}\right)$ SiC (solid, 25 °C) + $\left(\frac{b}{2}-\frac{d}{4}\right)$ SiO_2_ (cristobalite, 25 °C) + $\left(c-a+\frac{b}{2}-\frac{d}{4}\right)$ C (solid, 25 °C) + $\frac{d}{2}\;$ H_2_O (liquid, 25 °C)→Si_a_O_b-d/2_C_c_ + $\frac{d}{2}\;$ H_2_O (solid, 25 °C)	∆H^0^_f,comp_ (kJ/g-at.) = ∆H_1_ − $\left(a-\frac{b}{2}+\frac{d}{4}\right)\;$ ∆H_2_ − $\left(\frac{b}{2}-\frac{d}{4}\right)$ ∆H_3_ – $\frac{d}{2}$∆H_4_

**Table 6 materials-14-04075-t006:** Enthalpies of drop solution, enthalpies of oxidation at room temperature, enthalpies of formations from the elements and enthalpies of formation from the components for SiOC(H) samples.

Sample	Si_a_O_b_C_c_H_d_ (a + b + c + d = 1)				Δ*H_ds_* (kJ/g·at)	Δ*H_ox_* (kJ/g·at)	Δ*H_f_*_,*ele.*_ (kJ/g·at)	Δ*H_f_*_,*comp.*_ (kJ/g·at)
	a	b	c	d				
HQ4_SG 800 °C	0.286	0.619	0.047	0.048	−11.0 ±0.2	−28.0 ±0.2	−257.2 ±0.6	20.0 ±0.9
HQ4_SG 1100 °C	0.283	0.588	0.078	0.052	−12.3 ±0.2	−29.2 ±0.2	−265.8 ±0.6	−3.2 ±0.9
PhSiO_1.5__SG 800 °C	0.112	0.227	0.470	0.191	−203.8 ±3.3	−221.0 ±3.3	−93.2 ±3.3	−2.5 ±3.4
PhSiO_1.5__SG 1100 °C	0.142	0.230	0.568	0.060	−229.5 ±3.9	−244.7 ±3.9	−116.4 ±3.9	−13.9 ±3.9

## Data Availability

The data underlying this article will be shared on reasonable request from the corresponding author.

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
