# Peer review of "Structure and Thermodynamics of Silicon Oxycarbide Polymer-Derived Ceramics with and without Mixed-Bonding [Author-notes fn1-materials-14-04075]"

_materials, 2021, doi:10.3390/ma14154075_

Round 1
Reviewer 1 Report
Dear Authors,
I included my comments and suggestions and concerns for the manuscript with the following name submitted for Journal Materials, “Structure and Thermodynamics of Silicon Oxycarbide Polymer-derived Ceramics with and without Mixed-Bonding”
- The paper addresses on the novel synthesis of Silicon oxycarbides synthesized through a conventional polymeric route (PDC) with two preceramic polymers possessing unique organic groups.
- Authors should consider the broad range of the readers in the Journal of Materials Research.
- Provide a simple explanation for the concept provided in the following sentence “SiOC PDCs are X-ray amorphous up to temperatures of approximately 1300°C, making the structure and energetics of the amorphous state very interesting. Despite being X-ray amorphous, PDCs contain distinct domains at the nanoscale, making such microstructure largely responsible for thermodynamic and kinetic stability, properties that contribute to the materials’ resistance to crystallization and oxidation”.
- Please include what is meant by X-ray amorphous?
- Please delete the repeated word, amorphous.
- Provide a simple explanation for the concept provided in the following sentence “SiOC PDCs are X-ray amorphous up to temperatures of approximately 1300°C, making the structure and energetics of the amorphous state very interesting. Despite being X-ray amorphous, PDCs contain distinct domains at the nanoscale, making such microstructure largely responsible for thermodynamic and kinetic stability, properties that contribute to the materials’ resistance to crystallization and oxidation”.
- What kind of domains are they? Magnetic, electrical, grain, connected porosity? Are those domains defined as defect or not?
- Why to obtain materials with a property of being resistive to crystallization is desired? What is the advantage of it?
- All experimental information must have been collected under the experimental section. The following paragraph does not actually fit where it is currently.
- “Solid state magic angle spinning nuclear magnetic resonance (MAS NMR) provides a powerful tool for determining local bonding and heterogeneity present at the na-noscale16,11,14. Information about the presence of mixed bonding (SiOxC4-x) that occurs at the interfacial region between the silica rich region and the "free carbon" region is gained from both 13C and 29Si MAS NMR. Furthermore, Raman spectroscopy helps to identify the sp2 hybridized carbon species in the “free carbon” phase. The local nanostructural characteristics of PDCs are investigated by high-resolution transmission electron microscopy”.
- Once again please include a sentence for why this Silicon Oxycarbide Polymer-derived Ceramics are promising materials for industry, please briefly touch and provide answers for the following questions.
- Which industrial fields? What would be potential utilization area?
- What is the reason for their stability at high temperatures, what makes them different from crystalline counterparts?
- I could not find the name of the techniques used for establishing Table 1.
- “Analysis of the full elemental composition was determined by Mikroanalytisches Labor Pascher in Germany and is tabulated in Table 1, along with the corresponding atomic percentages in parenthesis.”
- Do authors have any comment on the chemical status, stoichiometry on the surface region (1-10 nm)?
- Are there oxygen functional groups on the surface? Such as any existence of oxygen containing groups? Chemisorbed species in particular?
- For the Figure 10;
- Is the mesoporosity distinct difference between two samples?
- Figure-10, insets on -b and –d: the images called inverse Fast-Fourier-Transform (Inv-FFT) images.
- What is the difference in between two micro-nano structure presented in those images, Figure 10, insets -b and –d?
- I really cannot observe the following “Even on a 2 nm scale, as given in the inset in d), no porosity was detected, while in b) the observed pore size ranges between 2-5 nm in diameter.”, please show this on the image for the reader.
- Are not those images magnified version of the HR-TEM images?
- What is the difference in between two micro-nano structure presented in those images, Figure 10, insets -b and –d?
- Are we looking at domains mentioned previously
- or cluster of atoms of Si-O-C? then where are the carbons ?
- Table 4, is oxygen molecule stabile at the temperature of 800°C?
Author Response
The response to reviewer's comments was included in the following attachment.

Reviewer 2 Report
Dear authors, I appreciate the effort made by you to carry out this study and to transpose it in the form of the paper submitted for review.
The experiment is presented coherently, exhaustively, and the explanations offered at each stage are consistent and clarifying.
As an observation for word editing: in the abstract PDC must be explained, and on page 18 the reference in the text is for table 5, but the table number on page is 4 (must be corrected).
Also, I consider that it would be worth specifying the percentages of the components of the mixtures you used at the Synthesis: water and TMOS, respectively methanol and TMOPS (page 3, lines 15 - 16).
One last addition that I consider important: you got 4 xerogels (HQ0.1_SG, HQ4_SG, HQ8_SG and PhSiO1.5_SG,), but you only experienced 2 - why?
Author Response

(The authors gave the same response as above.)

Reviewer 3 Report
Polymer-based ceramics (PDCS) are a fairly new class of ceramics that can be synthesized by crosslinking and pyrolysis of suitable polymer precursors. Recently, PDCS have attracted increased attention of researchers due to their outstanding properties, which include: stability with respect to decomposition and crystallization processes, as well as resistance to oxidizing and corrosive environments at high temperatures. In an article by the authors Casey Sugie et al. the synthesis of two pre-ceramic polymers with both phenyl substituents as unique organic groups is presented. A description of the synthesis method and a detailed analysis of the obtained materials are presented. It is shown that the mixed bond contributes to the thermodynamic stability of SiOC PDCs. The material of the article contributes to the understanding of the processes of synthesis of silicon oxycarbide and may be of interest to the readers of Materials. A note to the text is as follows : it is known that the solid-state structure of PDCS is significantly influenced by the molecular structure of polymer precursors, their composition and chemical bond. However, the influence of parameters such as the pyrolysis temperature and the heating rate are also important in the final structure of ceramics, which is not discussed in the article.
Author Response

(The authors gave the same response as above.)
